# Antibacterial Activity of Pharmaceutical-Grade Rose Bengal: An Application of a Synthetic Dye in Antibacterial Therapies

**DOI:** 10.3390/molecules27010322

**Published:** 2022-01-05

**Authors:** Michio Kurosu, Katsuhiko Mitachi, Junshu Yang, Edward V. Pershing, Bruce D. Horowitz, Eric A. Wachter, John W. Lacey, Yinduo Ji, Dominic J. Rodrigues

**Affiliations:** 1Department of Pharmaceutical Sciences, College of Pharmacy, University of Tennessee Health Science Center, 881 Madison Avenue, Memphis, TN 38163, USA; kmitachi@uthsc.edu; 2Department of Veterinary and Biomedical Sciences, University of Minnesota, 205 VSB, 1971 Commonwealth Avenue, Saint Paul, MN 55108, USA; yang1181@umn.edu (J.Y.); jixxx002@umn.edu (Y.J.); 3Provectus Biopharmaceuticals, Inc., 10025 Investment Drive, Suite 250, Knoxville, TN 37932, USA; epershing@pershingenterprises.com (E.V.P.); bhorowitz@pvct.com (B.D.H.); wachter@pvct.com (E.A.W.); jlacey@utmck.edu (J.W.L.III); rodrigues@pvct.com (D.J.R.)

**Keywords:** rose bengal (RB), high-purity form of rose bengal (HP-RB), antibacterial activity, drug-resistant gram-positive pathogens, multidrug-resistant bacteria, methicillin-resistant *Staphylococcus aureus*, vancomycin resistant *Enterococcus faecium*, biofilms, whole genome analyses

## Abstract

Rose bengal has been used in the diagnosis of ophthalmic disorders and liver function, and has been studied for the treatment of solid tumor cancers. To date, the antibacterial activity of rose bengal has been sporadically reported; however, these data have been generated with a commercial grade of rose bengal, which contains major uncontrolled impurities generated by the manufacturing process (80–95% dye content). A high-purity form of rose bengal formulation (HP-RBf, >99.5% dye content) kills a battery of Gram-positive bacteria, including drug-resistant strains at low concentrations (0.01–3.13 μg/mL) under fluorescent, LED, and natural light in a few minutes. Significantly, HP-RBf effectively eradicates Gram-positive bacterial biofilms. The frequency that Gram-positive bacteria spontaneously developed resistance to HP-RB is extremely low (less than 1 × 10^−13^). Toxicity data obtained through our research programs indicate that HP-RB is feasible as an anti-infective drug for the treatment of skin and soft tissue infections (SSTIs) involving multidrug-resistant (MDR) microbial invasion of the skin, and for eradicating biofilms. This article summarizes the antibacterial activity of pharmaceutical-grade rose bengal, HP-RB, against Gram-positive bacteria, its cytotoxicity against skin cells under illumination conditions, and mechanistic insights into rose bengal’s bactericidal activity under dark conditions.

## 1. Introduction

The increasing emergence of multidrug-resistant (MDR) Gram-positive bacteria is one of the major public health threats [1,2,3]. Particularly, MDR strains of Staphylococcus, Enterococcus, and Streptococcus spp. have a significant impact on morbidity and mortality [4]. The increasing resistance rates of these pathogens against critically important antibacterial agents (e.g., β-lactams, macrolides, aminoglycoside, fluoroquinolones, glycopeptide, oxazolidinones, cyclic peptides, and depsipeptides) are of great concern [5]. To date, very few new chemical entities have been examined in late-stage clinical studies for the treatment of infections caused by MDR bacterial pathogens [6]. Naturally occurring and synthetic dyes have been applied as antibacterial or antiprotozoal agents [7]. For example, methylene blue and clofazimine are still considered to be important orphan drugs [8,9]. Rose bengal (RB) dye (4,5,6,7-tetrachloro-2′,4′,5′,7′-tetraiodofluorescein) has been clinically investigated for the treatment of melanoma and other solid cancers [10,11,12,13,14]. Photodynamic antibacterial properties of RB have been reported for a limited number of bacteria [15,16,17,18,19,20,21,22,23,24,25,26,27]. RB is a bright rose-red xanthene compound that was first synthesized in the 19th century as a wool dye and subsequently used as a food dye in Japan (food red no. 105) [28]. The use of RB for the visual diagnosis of human ocular surface damage (via ocular instillation) was first described in 1914 (Feenstra et al. 1992) [29]. RB was later introduced as a diagnostic agent to evaluate the functional capacity of a human liver after a single 100 mg dose (Delprat et al. 1924) [30]. In 1971, ^131^I RB (Robengatope^®^, rose bengal sodium ^131^I injection USP) was approved by the U.S. Food and Drug Administration (FDA) for use as a diagnostic aid in determining liver function [31,32,33]. Commercial-grade RB, with purity varies between 80 and 95% RB, which includes gross contaminants and substance-related impurities, is manufactured using an historical process developed by Gnehm in the 1880s. It is assumed that RB used in diagnostic applications is a commercial-grade RB that contains some impurities [34]. The United States Pharmacopeia (USP) previously listed RB as analytical standards. RB was removed from the USP in 2019. Thus, commercial-grade RB lacks relevance in the context of modern diagnostic and therapeutic settings. Therefore, it poses significant regulatory challenges to validate RB for applications in the treatment of human diseases.

Provectus Biopharmaceuticals, Inc. (Provectus) realized several challenges in the purifications of RB; commercial-grade RB includes several transhalogenated substances (impurities derived from inherent side reactions occurring during the dye manufacturing process), as well as other by-products which have lost one or more iodides. Provectus concluded that commercial-grade RB is not capable of efficiently yielding a pharmaceutical-grade material with sufficient quantity to support clinical development and registration by the FDA and other global drug regulatory agencies. Provectus has established a novel multi-step approach for synthesizing and manufacturing RB [35,36]. Provectus’ synthesis and purification methods have been applied to a good manufacturing practice (GMP), producing RB to a pharmaceutical grade. A high purity RB (HP-RB) is manufactured under the guidelines of The International Council for Harmonization of Technical Requirements for Pharmaceuticals for Human Use (ICH), and designed to be applied as an injectable pharmaceutical or in topical medications [36,37,38,39]. Here, we wish to report the antibacterial activity of HP-RB and its scope and limitations as an antibacterial agent for the treatment of Gram-positive bacterial infections.

## 2. Results and Discussion

### 2.1. Antibacterial Activity of RB

The antibacterial activity of RB via photodynamic approaches has been studied in several research groups [15,16,17,18,19,20,21,22,23,24,25,26,27]. Applications of photodynamic therapy of RB are not limited to skin infections, including cellulitis, erysipelas, impetigo, folliculitis, and furuncles and carbuncles. RB can be immobilized on polymer supports that could successfully be applied for the eradication of bacteria on material surfaces and in water [17,18]. However, the spectrum of activity, rate of killing, and biofilm eradication activity of RB have been examined against specific bacteria. We have reinvestigated the bactericidal activity of pharmaceutical-grade RB (HP-RB) against a battery of Gram-positive and -negative bacteria, including *Mycobacterium* spp., under different light sources (fluorescence, LED, and sun lights) and dark conditions. The fluorescent light used was 17 W (1647 lumens, 63.8 cm^2^) and the LED was a 9.5 W (800 lumens, 28.3 cm^2^). In the experiments under the sunlight, through an architectural window, the 96-well plates were placed on the east side of the building (the BSL-2 lab on the 5th floor, College of Pharmacy, UTHSC) and the growth inhibition experiments were performed between 8 AM and 5 PM (on 18 June 2021). Minimum inhibitory concentrations (MICs, μg/mL), obtained via broth dilution and agar dilution methods (24h under fluorescent (23.0 KJ/cm^2^), and under LED light (29.0 KJ/cm^2^) conditions, and 9 h under sunlight) are summarized in Table 1 and Table 2. In these experiments, a series of FDA-approved drugs (meropenem, colistin, amikacin, linezolid, rifampicin, ciprofloxacin, azithromycin, isoniazid, capreomycin, and 5-fluorouracil) and preclinical antibacterial agents (tunicamycin, APPB) were included as the positive and negative controls of each MIC test [40,41,42,43,44]. RB used in Table 1 and Table 2 was a formulated product (HP-RBf, 10% RB in saline, Figure 1) [35,36]. HP-RBf effectively killed a wide range of Gram-positive bacteria with the MIC level of 0.20–3.1 μg/mL under illumination conditions (entries 1–23 in Table 1). The bactericidal activity of HP-RBf observed in entries 1–23 (Table 1) was not noticeably different depending on the light sources; the MIC values were equal or very close for fluorescent and LED lights. The HP-RB formulation killed Gram-positive *Bacillus* spp. at 0.39–0.78 μg/mL concentrations (entries 1–3). An MIC standard strain of *Staphylococcus aureus* displayed less susceptibility to HP-RBf; it required 1.6 μg/mL of HP-RBf to kill >99% of bacteria (entry 4). HP-RBf’s bactericidal activity was examined against a panel of seven methicillin-resistant *S. aureus* (MRSA) with different SCCmec types (entries 5–11) [45]. Under the dark condition, HP-RBf showed antibacterial activity against all Gram-positive bacteria listed in entries 1–23 (Table 1) at 25.0–100 μg/mL concentrations. Under dark conditions, commercial RB is known to display antibacterial activity at high concentrations. Our data support that RB has unknown mechanisms to inhibit the growth of bacteria other than through the excitation mechanism of triplet oxygen, which generates cytotoxic reactive oxygen species. While excellent antibacterial activities were observed against Gram-positive bacteria, three *E. coli*, two *Pseudomonas aeruginosa*, two *Klebsiella pneumoniae*, and two *Acinetobacter baumannii* strains showed resistance to HP-RBf; the MIC levels were 50 or >100 μg/mL against these Gram-negative bacteria (entries 31–39). All MRSA strains tested in Table 1 were killed by HP-RBf at 0.78–3.1 μg/mL concentrations under the fluorescent or LED light. HP-RBf was further examined against four vancomycin-resistant *S. aureus* strains (entries 12–16); all vancomycin-resistant strains were killed at below 1.0 μg/mL concentrations. *Staphylococcus epidermidis* is also effectively killed by HP-RBf under an aerobic condition (entry 16). Drug-susceptible and -resistant *Enterococcus faecalis,* including vancomycin-resistant strains, were killed at a concentration range of 0.39–0.78 μg/mL of HP-RBf (entries 17–21). *Streptococcus salivarius* was susceptible to HP-RBf (entry 23), whereas *Streptococcus pneumoniae* showed resistance to HP-RBf (entry 24).

An anaerobic Gram-negative bacterium, *Bacteroides fragilis,* tolerated the HP-RBf treatment at 100 μg/mL or higher concentrations. Although the efficacy of RB against a large group of Gram-negative bacteria has not thoroughly been investigated, a photodynamic approach using RB in the presence or absence of KI was studied to inhibit the growth of *Salmonella* and *Burkholderia* spp. Our data, summarized in entries 41–42 (Table 1), suggest that HP-RBf is effective in killing *Burkholderia, Salmonella*, and *Proteus* spp. at 3.13–12.5 μg/mL concentrations under fluorescent or LED lights. All Gram-negative bacteria tested in Table 1 were not susceptible to HP-RBf under the dark condition; the MICs were determined to be >100 μg/mL (entries 31–47). The antibacterial activity of HP-RBf against 5 *Mycobacterium* spp. was examined; the MIC values of HP-RBf under the illuminated conditions were 12.5–25.0 μg/mL (entries 25–29), which were 15–60-fold higher than those for the Gram-positive bacteria (entries 1–23). Interestingly, under the dark condition, these Mycobacteria were killed at an equal or similar MIC to those observed under illumination conditions. We have examined the MIC of HP-RBf against a limited number of bacteria under sunlight (filtered through glass window). The MIC values for eight Gram-positive bacteria and three Gram-negative bacteria displayed good agreement with those obtained under fluorescent and LED lights (entries 1–4, 16, 21–23, 43 and 47). We also examined the bactericidal effects of HP-RBf against a yeast strain; HP-RBf inhibited the growth of *Saccharomyces cerevisiae* at the same MIC level (12.5 μg/mL) under the three different light sources and at much higher concentrations under the dark condition (125 μg/mL) (entry 30).

We observed that under the illumination conditions, the MIC values of HP-RBf that were determined by the agar dilution method were lower than those determined by the broth dilution method (Table 1). Selected examples of difference in the MIC values determined by the two methods are summarized in Table 2. The growth of Gram-positive bacteria, such as *B. subtilis*, *B. cereus*, and *S. aureus* were inhibited at 0.01–0.10 μg/mL concentrations (entries 1–4 in Table 2), which were 7~70-fold less than the MIC values determined by the broth dilution method. *E. coli* (35218^TM^) and *B. cepacia* (UCB717) strains were also far more susceptible to HP-RBf under fluorescent light in the agar dilution method than those in the broth dilution method (entries eight and nine). Similarly, *M. smegmatis* (ATCC607^TM^) was killed at lower concentrations on the drug-containing agar plates (or wells) than in those in broth (entry 10) [46]. Under the dark condition, the MICs of HP-RBf, determined via the agar dilution method, displayed good agreement with the values measured in the broth dilution method (entries one to seven). Minimum bactericidal concentrations (MBCs) of HP-RBf against the selected bacteria are also summarized in Table 2. HP-RBf has a cytostatic effect against a *S. cerevisiae* sp; it displayed a 50% growth inhibition at 1.6 μg/mL, but required 400 μg/mL (MBC) to kill >99% of the yeast under the fluorescent light. *S. cerevisiae* was not killed by 500 μg/mL of HP-RBf under the dark condition (entry 11).

The MIC data summarized in Table 1 and Table 2 indicate that HP-RBf has strong bactericidal activity against Gram-positive bacterial and a limited number of Gram-negative bacteria (some *E. coli* strain and *Burkholderia* spp.) under illuminated conditions. HP-RBf kills *Mycobacterium* spp. at relatively high concentrations (12.5–25.0 μg/mL).

### 2.2. Time-Kill Kinetics of RB

Photodynamic growth inhibitions of RB against several bacteria have been previously studied; Sabbahi et al. reported that, under a visible light exposure, ~80% of a *S. aureus* strain lost its viability in 10 min with 19.5 μg/mL of RB (a light fluence dose of 30 J/cm^2^) [26]. Considering the MIC values determined under the fluorescent light (24 h, 23.0 KJ/cm^2^, Table 1), we performed the time-kill kinetics assays of HP-RBf, with one drug susceptible strain of *S. aureus* 6538^TM^, as well as three drug-resistant Gram-positive bacterial strains (*S. aureus* BAA-44, *S. aureus* 71,080 (VRS8), and *E. faecium* NR-32065), and one Gram-negative bacterium (*B. cepacia* UCB717). Our preliminary studies suggested that HP-RBf kills both Gram-positive and -negative bacteria with a 6-log reduction within 2 h. Thus, the time-course experiments on the selected bacterial strains were conducted under the fluorescent light for 2 h (0–1130 J/cm^2^) at concentrations of 2~8-fold the MIC (HP-RBf). Reference molecules used were linezolid (10 μg/mL) and ciprofloxacin (10 μg/mL) for the Gram-positive bacteria and amikacin (10 μg/mL) and meropenem (10 μg/mL) for the Gram-negative bacterium. HP-RBf reduced 4.6 × 10^8^ (colony forming units: CFUs) of *S. aureus* 6538^TM^ by a log reduction of six in 1 min. No CFU was counted after 2 min at 1.6 and 5.0 μg/mL (HP-RBf) concentrations (Figure 2A). Similarly, HP-RBf killed *S. aureus* BAA-44, *S. aureus* 71,080 (VRS8), and *E. faecium* NR-32065 within 2 min at a concentration that was two times higher than the MIC of HP-RBf (Figure 2B). *B. cepacia* UCB717 (2.9 × 10^8^ CFU) was killed in concentration- and time-dependent manners by the treatment of HP-RBf (Figure 2C); at four times the MIC concentration, there was a >5 log reduction of the bacteria in 5 min. HP-RBf required 40 min to reduce a number of bacteria with a 5 log reduction at two times the MIC concentration. The fast-killing nature of HP-RBf confirmed in the selected case studies (Figure 2) has significant advantages over the approved antibacterial agents, in that (1) there was an increase of the safety profile in applications for disinfection and sterilization, and (2) lowering the frequency of generating drug-resistant strains.

### 2.3. Anti-Biofilm Activity of RB

The fast-killing antibacterial character of RB observed in Section 2.2. encouraged us to evaluate the anti-biofilm efficacy of HP-RBf in Gram-positive bacteria. The data, summarized above, indicate that HP-RBf possesses a significant drug affinity or permeability onto (or into) Gram-positive bacteria. Antimicrobial and antifungal photodynamic therapy have been studied with the photosensitizers under biofilm conditions; however, a limited number of bacterial biofilms have been examined with RB [15]. Recently, anti-biofilm activity of RB against cariogenic oral bacteria, harboring on the tooth surface, was demonstrated under blue light LED (Hirose et al. 2021) [20]. Here, we examined the efficacy of HP-RBf against biofilms of a drug-susceptible *S. aureus* 6508^TM^, and drug-resistant *S. aureus* 71,080 (VRS8) and *E. faecium* NR-32065 under the florescent light and dark conditions. Linezolid is not an effective drug in eradicating biofilms of Gram-positive bacteria but has a beneficial effect in the prevention of biofilm formations [47]. We applied linezolid as a positive control at very high concentration of 600 μg/mL (>100× MIC for the planktonic cells) in our biofilm assays. We have confirmed that all strains tested here form strong biofilms on the polystyrene well plates. Under the fluorescent light, HP-RBf could eradicate the biofilms of *S. aureus* 6508 with a 7-log reduction at 30.0 μg/mL (38× MIC) concentration, which demonstrated the same level of efficacy as that observed for linezolid (at 600 μg/mL) (Figure 3A). HP-RBf showed a biofilm eradication activity in a dose-dependent manner. At a 60.0 μg/mL (77× MIC) concentration, over a 5-log reduction was achieved. No CFUs were counted at 100 μg/mL. Although it required much higher concentrations, HP-RBf showed biofilm eradication activity under the dark condition; at 50.0 μg/mL (2× MIC under dark) concentration, HP-RBf significantly reduced the number of viable bacteria. At a 500 μg/mL (20× MIC) concentration, only 110~150 CFU/mL was observed. No viable bacteria appeared at 1000 μg/mL concentration. These trends could be observed in the biofilms of the drug-resistant *S. aureus* 71,080 (VRS8) and *E. faecium* NR-32065 with much lower HP-RBf concentrations (Figure 2B,C). Under the fluorescent light, over a 6 log reduction of viable bacteria was observed at a 10.0 μg/mL concentration. No CFUs were counted at 30.0 μg/mL or higher concentrations.

There is some debate as to whether *S. aureus* colonies can be considered as air-exposed biofilms [48]. Nonetheless, our studies have shown that, in the colonies of *S. aureus* grown on agar plates (at 37 °C for 2 days), it is not possible to reduce the number of viable cells with FDA-approved antibiotics in a few hours. Figure 4 and Table 3 summarizes the effect of HP-RBf on air-exposed biofilms of a MRSA, *S aureus* BAA-44^TM^, under fluorescent light (17 W, 1 h, 0.57 KJ/cm^2^). Spraying HP-RBf (5.0 μg/mL or 10 μg/mL solution, 250 μL (twice)) and fluorescent light exposure (for 1 h) eradicated viable bacteria in the biofilms with a 4 log reduction (determined at a dilution of 5.8 × 10^9^). The experiments, summarized in Figure 4, indicate that HP-RBf can readily permeate biofilm matrix and diffuse across Gram-positive bacterial cell walls. These observations strongly support that HP-RBf has the potential to treat serious bacterial skin infections.

### 2.4. Antimycobacterial Mechanisms of RB

The antibacterial photodynamic therapy of RB has been reported in several articles. Permeation of RB through bacterial cell walls and binding to cell membranes followed by the production of reactive oxygen species are likely bactericidal mechanisms in illumination conditions [49,50]. Although relatively high concentrations were required, HP-RBf killed a majority of Gram-positive bacteria, including *Mycobacterium* spp., in dark conditions. It also effectively eradicated biofilms of Gram-positive bacteria (Section 2.3). Antibacterial activity of RB in dark conditions remains far from completely understood [51,52]. HP-RBf killed *Mycobacterial* spp. at 12.5–25.0 μg/mL in illumination conditions, and at 25.0–50.0 μg/mL in dark conditions (Table 1). In both conditions, HP-RBf killed five *Mycobacterial* spp. at a slower rate than that of Gram-positive bacteria. HP-RBf seems to have a low permeability of mycobacterial cell walls. Due to the rapid bactericidal effect of HP-RBf against Gram-positive bacteria even in the dark condition, generation of RB-resistant mutants of Gram-positive bacteria is an extremely difficult task. We successfully generated RB-resistant mutants of *M. smegmatis*, which had the MIC value of 200 μg/mL [46]. The RB-resistant strain was susceptible to most TB drugs (amikacin, capreomycin, rifampicin, APPB, and ethionamide) (Table 4). However, it showed a cross-resistance to INH. INH is a prodrug that requires oxidative activation by KatG, which belongs to catalase–peroxidases. KatG oxidizes INH to form an electrophilic species, an isonicotinoyl radical molecule, which reacts with the NADH-dependent enoyl-ACP (acyl carrier protein) reductase, an enzyme involved in the biosynthesis of mycolic acids in mycobacteria (Figure 5A) [53,54]. The RB-resistant *M. smegmatis* strain acquired medium INH resistance but did not show resistance to ethionamide (ETH). The major mechanism of INH resistance is mutation in *katG*, while ETH is activated by the monooxygenase EthA [55]. Our observations may imply that the RB’s antimycobacterial mechanisms share one or more INH metabolic enzymes to form bactericidal species. To elucidate a potential mechanism of action, we performed a whole-genome sequencing analysis of an RB-resistant *M. smegmatis* ATCC607 strain, using the next-generation of DNA sequencing technologies [56]. We identified that one insertion mutation occurred in anti-sigma E factor gene (*rseA:* evidenced TG:104 vs. T:0) and the aquaporin family protein gene (evidenced by GCACCCT:71 vs. G:0), respectively. Consequently, these insertion mutations caused the reading frame changes in the corresponding proteins and generated truncated proteins when compared to its parental strain. It has been reported that RseA functions as a specific anti-sigma E factor in *Mycobacterium tuberculosis* and that the sigma E factor (SigE) enables mycobacterial organisms to tolerate a variety of stress responses [57,58]. Thus, the expression of a non-functional RseA in the RB-resistant mutant may affect the activity of SigE, increasing the bacterial tolerance to RB. On the other hand, the aquaporin family proteins exist in various organisms and play a critical role in the bidirectional flux of water and uncharged solutes cross cell membranes. It was reported that a null mutation of the *Streptococcal* aquaporin homolog increased the intracellular H_2_O_2_ retention, indicating that aquaporin mediates transporting H_2_O_2_ in *Streptococcal* spp. [59].

Therefore, we hypothesize that HP-RBf may inhibit the aquaporin function, leading to the accumulation of H_2_O_2_ within the bacterial cells. Interestingly, a single nucleotide deletion that caused the frameshift mutation was observed in molybdopterin-dependent oxidoreductase (evidenced T:74 vs. TC:0) of the RB-resistant strain. The oxidoreductase systems can form superoxide by the reduction of molecular oxygen, or NO by the reduction of inorganic nitrate. RB may serve as a single-electron acceptor in the redox of the oxidoreductases that will produce the radical anion (RB^•^-) or RB triplet state, undergoing the electron transfer reaction with oxygen. As such, we propose the involvement of molybdopterin-dependent oxidoreductase in the generation of reactive oxygen or nitrogen spp. through the excitation of RB under dark conditions (Figure 5B). The *katG* gene was intact in the RB resistant strain. Thus, it remains difficult to speculate a mechanism that confers the cross-resistance with INH remains. However, we observed mutations in several transcriptional regulators of the RB-resistant strain (see SI) that may affect the expression level of KatG, suppressing the INH activation. It generates hydroxy radicals (reactive oxygen species) through the Fenton reaction of H_2_O_2_. The requirement of relatively high concentrations of HP-RBf to display bactericidal activity against Gram-positive bacteria, including mycobacterial spp., may imply that the affinity of RB with catalases is moderate. Similarly, a cytotoxicity mechanism of RB in mammalian cells may be explained.

### 2.5. Cytotoxicity of HP-RBf under Lights

The cytotoxicity of HP-RBf has been extensively evaluated in Provectus’ oncology drug development program, where it was investigated via intralesional administration for the treatment of melanoma and hepatic tumors [37,38]. The results of Provectus’ toxicology studies showed that HP-RBf does not have systemic toxicological effects, mutagenic potential, nor female reproductive and development effects at therapeutic concentrations [60]. These data are described in Provectus’ U.S. patents (Eagle et al. 2019) [61]. Besides the toxicity studies in systemic applications, the cytotoxicity of pharmaceutical-grade RB (HP-RBf) against mammalian cells in illumination conditions has not been discussed.

RB has been used for over 50 years to diagnose eye and liver disorders. It is often useful as a stain in diagnosing certain medical issues, such as conjunctival and lid disorders (*vide supra*). In these applications, 0.1–2.0% RB has been used. RB in concentrations below 2.0% is considered to be safe under natural and artificial lights [62]. The cytotoxicity level of RB against healthy cells under illumination conditions should be clarified for photodynamic antibacterial chemotherapy; however, these data are not publicly available. We chose two healthy cell lines, Vero (the kidney of an African green monkey) cells and skin (human epidermal keratinocytes (HEKa) cells to determine the in vitro cytotoxicity of HP-RBf under fluorescent light. We have generated large data sets of the cytotoxicity of antibacterial and anticancer agents against Vero cells, which allows us to compare the toxicity level of new molecules [63,64]. The cytotoxicity against HEKa cells provides useful toxicology information for the development of safe, topically-applied antibacterial agents [65,66]. In this research program, HEKa cells were differentiated to a stratified squamous epithelium via an air–liquid interface; this type of epithelium can be applied as a physiological tissue to study epidermal necrolysis by the treatment of HP-RBf. In a 24 h experiment under dark conditions, HP-RBf showed the IC_50_ value of 300 μM (292 μg/mL) against Vero cells. Under the fluorescent light condition, HP-RBf displayed cytotoxicity against Vero cells in a time- and -concentration-dependent manner. Figure 6 summarizes the time effect of the confluent Vero cell from the treatment of HP-RBf (0–300 μM). Nearly 50% of confluency was lost by the treatment of a 100 μM (97.4 μg/mL) concentration of HP-RBf in 4 h; however, the monolayer was intact for 1 h exposure of 100 μM of HP-RBf. Around 18% of confluence was lost in 1 h at 200 μM, while over 80% of confluence was lost in 4 h. At 300 μM concentrations, complete loss of cell viability was observed within 2 h. Thus, it was concluded that, under fluorescent light, Vero cells are tolerated at 100–200 μM for 1 h. Time-kill kinetic studies, summarized in Section 2.2, indicated that HP-RBf kills Gram-positive bacteria in 1–2 min and Gram-negative bacteria in 5 min. The selectivity index (SI), a ratio that measures the window between cytotoxicity and antibacterial activity, was determined to be >62.5 (for Gram-positives) and >7.9 (for Gram-negatives) for an 1 h treatment time. These favorable toxicity profiles of HP-RBf were further supported by the cytotoxicity studies, using HEKa cells (*vide supra*). HP-RBf was localized in the stratum layers. HP-RBf did not cause necrosis of the stratum corneum cells at 10 and 100 μM at a 1 h exposure. Some necrosis was observed on the surface tissue when the concentration increased to 200 μM (Figure 7). Therapeutic concentrations of HP-RBf are likely to be between 5 and 10 μM; thus, these in vitro cytotoxicity tests imply that skin infections can be treated with HP-RBf without causing cytotoxicity of host cells under illumination conditions.

## 3. Materials and Methods

### 3.1. Formulation of Pharmacological Grade Rose Bengal in Saline (HP-RBf)

Rose bengal disodium salts were synthesized according to Provectus’ proprietary procedure. The detailed procedure was described previously [12].

### 3.2. Acquisition of Bacteria

The drug susceptible bacteria and yeasts used in this program were purchased from ATCC (The American Type Culture Collection). The drug resistant strains were acquired from BEI Resources (NIAID).

### 3.3. Log Phase Bacterial Culture

All liquid bacterial culturing was performed with a conical flask with an air filter. A single colony of a bacterial strain was grown, according to the conditions recommended by ATCC. Seed cultures and larger cultures of bacteria were obtained using media recommended by ATCC. *M. smegmatis* (ATCC607) was cultured on a 0.5% Tween 80 Middlebrook 7H10 nutrient agar (0.4% glycerol) [46]. The culture flasks were incubated for 3–4 days for *M. smegmatis* (ATCC607), and for 10–12 days for *M. tuberculosis* H_37_Rv in a shaking incubator at 37 °C, with a shaking speed of 200 rpm, and were cultured to the mid-log phase (optical density—0.5). The optical density was monitored at 600 nm using a 96-well microplate reader. Anaerobic bacteria were grown in an anaerobic chamber under an atmosphere of a mixture of H_2_ and N_2_ (5/95%) with a palladium catalyst.

### 3.4. MIC Assays

All testing followed the guidelines set by the Clinical & Laboratory Standards Institute (CLSI) [68]. Minimum inhibitory concentrations (MICs) were determined by broth dilution microplate alamar blue assay or by OD measurement. All compounds were stored in DMSO or saline (1 mg/100 μL concentration). This concentration was used as the stock solution for all MIC studies. Each compound from stock solution was placed in the first well of a sterile 96-well plate and a serial dilution was conducted with the culturing broth (total volume of 10 μL). The bacterial suspension at log phase (190 μL) was added to each well (total volume of 200 μL), and was incubated for 24 h (2–14 days for Mycobacterium spp.) at 37 °C (27 °C for yeasts). 20 μL of resazurin (0.02%) was added to each well and incubated for 4 h (National Committee for Clinical Laboratory Standards (NCCLS) method (pink = growth, blue = no visible growth)). The OD measurements were performed for all experiments prior to performing colorimetric assays. The absorbance of each well was also measured at 570 and 600 nm via UV-Vis.

The MIC values were also determined by using drug-containing agar plates. The bacterial culture of 1 × 10^5^ and 1 × 10^9^ CFU/mL was plated and incubated at the appropriate temperature (37 °C for bacteria, 27 °C for yeasts) and duration. The MIC valued were determined via counting the colony-forming units (CFUs), for the concentrations that caused a 3-log-fold decrease in CFU/mL (see Section 3.5).

### 3.5. Minimal Bactericidal Concentration (MBC) Assays

A single colony of a specific bacterium (grown on an agar plate) was inoculated into the culture broth. A bacterial culture was grown overnight, then diluted in growth-supporting broth to a concentration between 1 × 10^5^ and 1 × 10^9^ CFU/mL. Based on the MIC values, agar plates containing a drug (MIC, 2×–20× MIC) were prepared. A series of drug-containing agar plates were inoculated with equal volumes of the specific bacterium. The agar plates were incubated at the appropriate temperature and duration. CFU/mL were counted. The MBC values were determined by reduction of >99.9% of bacteria [46,64].

### 3.6. Time-Kill Kinetic Assays

A time-kill kinetics assay for antimicrobial agents was performed using the CLSI guidelines, with a minor modification. The multiple time points in the time-kill kinetics assays for HP-RBf and the reference molecules were performed. The bacterial culture grown in the broth was diluted to a concentration between 1 × 10^8^ and 5.0 × 10^9^ CFU/mL. A stock dilution of the antimicrobial test substance was prepared at approximately 2~8-fold of the MIC values. The test compounds were inoculated with equal volumes of the specified bacteria, placed in a 96-well plate. The microtiter plates were incubated at 37 °C and for various durations (1–120 min) under fluorescent light (conditions are summarized in the figure legend). An aliquot of the culture media was taken from each well and a serial dilution was performed. The diluted culture was incubated at 37 °C and the CFU/mL was counted. Bactericidal activity was defined as a greater than 3 log-fold decrease in colony forming units [64].

### 3.7. Anti-Biofilm Assays

The biofilms were generated on 12-well plates by incubating each bacterium for five days. The planktonic bacteria in the culture media were gently removed, and fresh media were placed. HP-RBf (25× MIC) or linezolid (200× MIC) were added into each well and incubated at 37 °C under dark or light for 24 h (6.7 KJ/cm^2^). An aliquot of each culture was diluted (×10,000 or ×100,000) on agar plates at 37 °C for 24 h. CFU/mL was counted [47,69].

The bacteria (−1.0 × 10^4^) were spread on an agar plate and incubated in an oven at 37 °C for 2 days. The air-exposed biofilms on an agar plate were treated with HP-RBF. After 1 h under fluorescent light (0.57 KJ/cm^2^), saline (2 mL) was added to the agar surface. The bacterial suspension (100 μL) was taken into a sterile tube, and serial dilutions were performed. The diluted sample (100 μL) was spread on agar plates and incubated at 37 °C for 24 h. CFU/mL was counted.

### 3.8. Mammalian Cell Lines and Culturing

Vero cells (ATCC CCL-81) were purchased from the ATCC. HEK cells (SCCE020) were purchased from MilliporeSigma. The cell lines were cultured and maintained in the media as recommended by the suppliers. 

Vero cells were cultured in Eagle’s Minimum Essential Medium (MEM) (CORNING, 10-009-CV) supplemented with 10% FBS, 1% Penicillin–Streptomycin Solution (cellgro, 30-002-CI), 1% HyClone MEM Non-Essential Amino Acids Solution (100×) (GE Healthcare Life Sciences, SH30238.01), and 1% HyClone Sodium Pyruvate 100 mM Solution, (GE Healthcare Life Sciences, SH30239.01), and incubated in a 37 °C incubator with 100% humidity and 5% CO_2_. This was refreshed with fresh medium every 2 days until the culture reached 100% confluence, which takes approximately 5 days depending on the proliferation rate.

Human Epidermal Keratinocytes, Neonatal (MILLIPORE, Catalog# SCCE020) were cultured in EpiGRO™ Human Epidermal Keratinocyte Complete Culture Media Kit (MILLIPORE, SCMK001) and incubated in a 37 °C incubator with 100% humidity and 5% CO_2_. This was refreshed with fresh medium after 2 days and three times weekly until the culture reached 100% confluence.

### 3.9. Cytotoxicity Assay with Vero Cells

All testing followed the guidelines set by the Clinical & Laboratory Standards Institute (CLSI), with minor modifications. Cytotoxicity assays for HP-RBf were performed in a 24-well plate. Into each well (1 mL medium/well), 1 μL of each drug concentration was added. After 1, 2, 3, and 4 h of incubation under the light at r.t., the medium was removed and the cell was washed with PBS (x3). After adding the medium (1 mL/well), 10 μL of MTT solution (5 mg/mL in PBS) was added and incubated for another 3 h at 37 °C (5% CO_2_). The medium was removed, and DMSO (1 mL/well) was added. Viability was assessed on the basis of cellular conversion of MTT into a purple formazan product. The absorbance of the colored formazan product was measured at 570 nm by a BioTek Synergy HT Spectrophotometer [43].

Vero cells (5 × 10^4^ cells/well (in 196 μL of the culture medium)) were plated in a 96-well plate and the cell cultures were incubated for 4 days to form the monolayer (100% confluence). Into each well, HP-RBf (0–300 μM) was added. Images were obtained every hour using an IncuCyte Live-Cell Imaging System (Essen BioScience, Ann Arbor, MI). Cell proliferation was quantified using the metric phase object confluence (POC), a measurement of the area of the field of view that is covered by cells, which is calculated by the integrated software [70].

### 3.10. Cytotoxicity Assay with HEKa Cells

We referenced the protocols described in Testing Cell Monolayer Integrity on Transwell Permeable Supports (CLS-AN-047W). Cytotoxicity of HP-RBf against Human Epidermal Keratinocytes (HEKa) was evaluated in a hanging cell culture insert in a 24-well plate. HEKa (4 × 10^5^ cells/mL, 0.5 mL) cell suspension) was placed into the insert where the outside of the inserts was filled with EpiGRO™ Complete Culture Media (1.5 mL) [67]. This was incubated at 37 °C overnight. The next day, without removing the original medium, an additional 0.5 mL of the media was added to the inside of each insert and incubated at 37 °C for 2 days. High quality HEKa approached 100% confluence by the third day of submerged culture. On the 4th day, the media from each insert was gently aspirated. The HEKa cell cultures were maintained at the air/liquid interface. The HEKa culture was incubated at 37 °C for an additional 10 days. During the 10 day incubation, the media was changed every other day and any bubbles were removed from beneath the insert membrane. After 25 days of 3D HEKa culture, approximately 6 to 8 layers of live epithelium were produced. HEKa tissues were treated with HP-RBf (0–200 μM) for 1 h under fluorescent light at r.t., washed with PBS (×3), and fixed with 4% formalin for 1 h at r.t. The HEKa tissues were released from the insert and embedded into paraffin. Sections of 4 μm were cut and transferred onto slides for hematoxylin and eosin (H&E) staining [71].

### 3.11. Whole-Genome Sequencing of M. smegmatis Strains

RB-resistant *M. smegmatis* (ATCC607^TM^) strains were generated according to the procedures reported previously [72]. To identify single nucleotide polymorphisms (SNPs) that may contribute to the bacterial resistance to HP-RBf, the genomic DNAs were purified from the stationary cultures of an HP-RBf-resistant mutant and its parental control, *M. smegmatis* 607^TM^, according to the procedure reported previously [46]. The purified genomic DNA was submitted to the University of Minnesota Genomic Center (UMGC) for quality control analysis, and the library preparation and DNA sequencing was performed using an advanced Illumina MiSeq DNA-seq technology. Sequence reads from the mutant and the control were evaluated for their quality using FastQC. Low-quality tails and adapters were removed with Trimmomatic [73]. The whole-genome sequence of the *M. smegmatis* strain FDAARGOS_679 was used as a reference, and SNPs or other variants such as deletion and insertions were called by using a bioinformatic tool Snippy (https://github.com/tseemann/snippy (accessed on 10 October 2021)) (see Appendix A).

## 4. Conclusions

Provectus has established a manufacturing and purification process for pharmaceutical-grade RB that fulfills both cGMP and ICH requirements. We have evaluated the antibacterial activity and cytotoxicity of a pharmaceutical-grade RB formulated product (HP-RBf) in illuminated and dark conditions. The comprehensive MIC data for HP-RBf summarized here indicate that HP-RBf is very effective in killing most Gram-positive bacteria (MIC 0.39–3.1 μg/mL), except for *Streptococcus pneumoniae* spp. *S. pneumonia* is one of very few Gram-positive bacteria that is susceptible to colistin (polymixin E), an anti-Gram-negative drug. HP-RBf kills *Mycobacterial* spp. with the MIC values of 12.5–25.0 μg/mL under illumination conditions. It is speculated that the mycolic acid-containing thick cell walls reduce the cellular uptake of the charged RB, increasing its MIC values against *Mycobacterial* spp. much higher than those of Gram-positive bacteria. We confirm that HP-RBf is an excellent agent to eradicate the biofilms of Gram-positive bacteria, including drug-resistant strains. Under fluorescent and dark conditions, HP-RBf significantly reduced the number of viable cells of the drug-resistant strains of *S. aureus* and *E. faecium* in a concentration-dependent manner. HP-RBf displayed a significant bactericidal effect on the air-exposed biofilms of a MRSA strain under fluorescent light exposure within 1 h. These studies indicate that RB has the potential to be used as an antibacterial agent to kill drug-resistant bacteria in any growth phase. 

RB is an anionic photosensitizer with high singlet oxygen quantum yield. This is its primary mode of action of antibacterial activity in illumination conditions. We could successfully generate RB-resistant mutants of *M. smegmatis* under the dark condition. It showed a cross-resistance to a first-line TB drug, INH. We performed whole-genome analyses of the generated resistant mutant. It revealed the unique mutations that may confer mechanisms of HP-RBf’s bactericidal activity. Several redox systems, transcriptional factors, and aquaporin may be responsible for HP-RBf’s resistance mechanisms. Based on these data, it is speculated that HP-RBf can be shifted to an excited state by the enzymes associated with the oxidoreductases, forming reactive oxygen or nitrogen spp. under dark conditions. Certain cooperative mechanisms may exist in the bactericidal activity of HP-RBf under dark conditions. 

To summarize, we demonstrated that a pharmaceutical-grade formulation of RB, HP-RBf, has an appropriate selectivity index for the treatment of Gram-positive bacterial infections under illumination conditions. The rapid bactericidal effect of HP-RBf that is effective against bacterial biofilms is an unusual drug characteristic and promises the advancement of HP-RBf as an antiseptic for clinical applications.

## Figures and Tables

**Figure 1 molecules-27-00322-f001:**
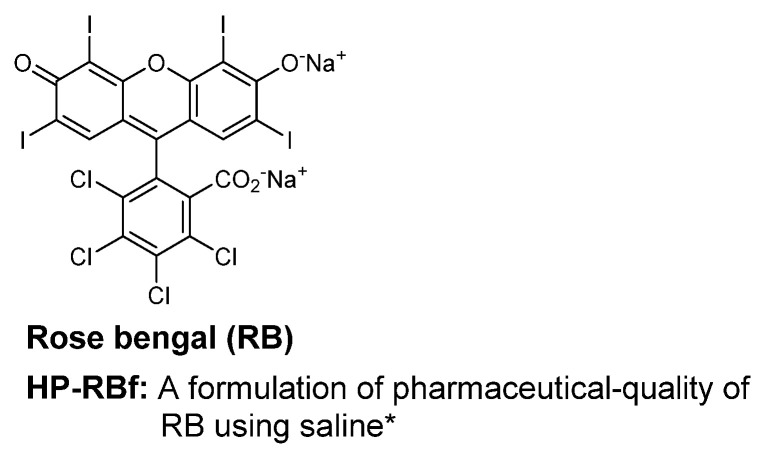
Rose bengal (RB) with >99% purity. * A 10% concentration of pure RB disodium salts in saline solution was applied in this article.

**Figure 2 molecules-27-00322-f002:**
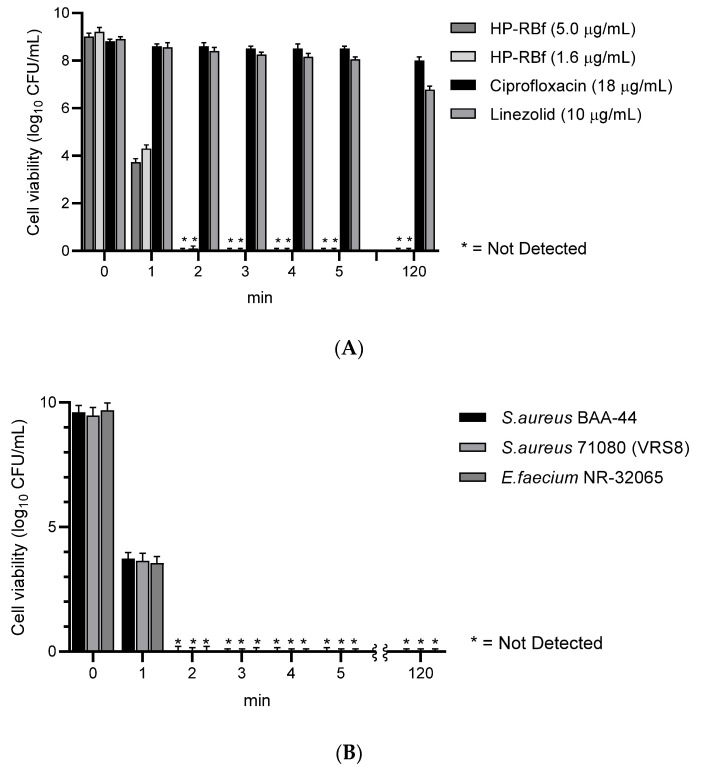
Time-kill kinetics of HP-RBf (10% RB in saline) against Gram-positive and -negative bacteria under the fluorescent light (17 W, 63.8 cm^2^, 0–1.1 KJ/cm^2^).^a^ (**A**) Time-kill kinetics of HP-RBf and representative antibiotics against *S. aureus* 6538TM; (**B**) Time-kill kinetics of HP-RBf against drug resistant Gram-positive bacteria; (**C**) Time-kill kinetics of HP-RBf and anti-Gram-negative antibiotics against B. cepacia (UCB707). ^a^ Two times the MIC concentration was applied: 3.2 μg/mL for *S. aureus* BAA-44; 0.8 μg/mL for S. aureus 71,080 (VRS10); 1.6 μg/mL for *E. faecium* NR-32065.

**Figure 3 molecules-27-00322-f003:**
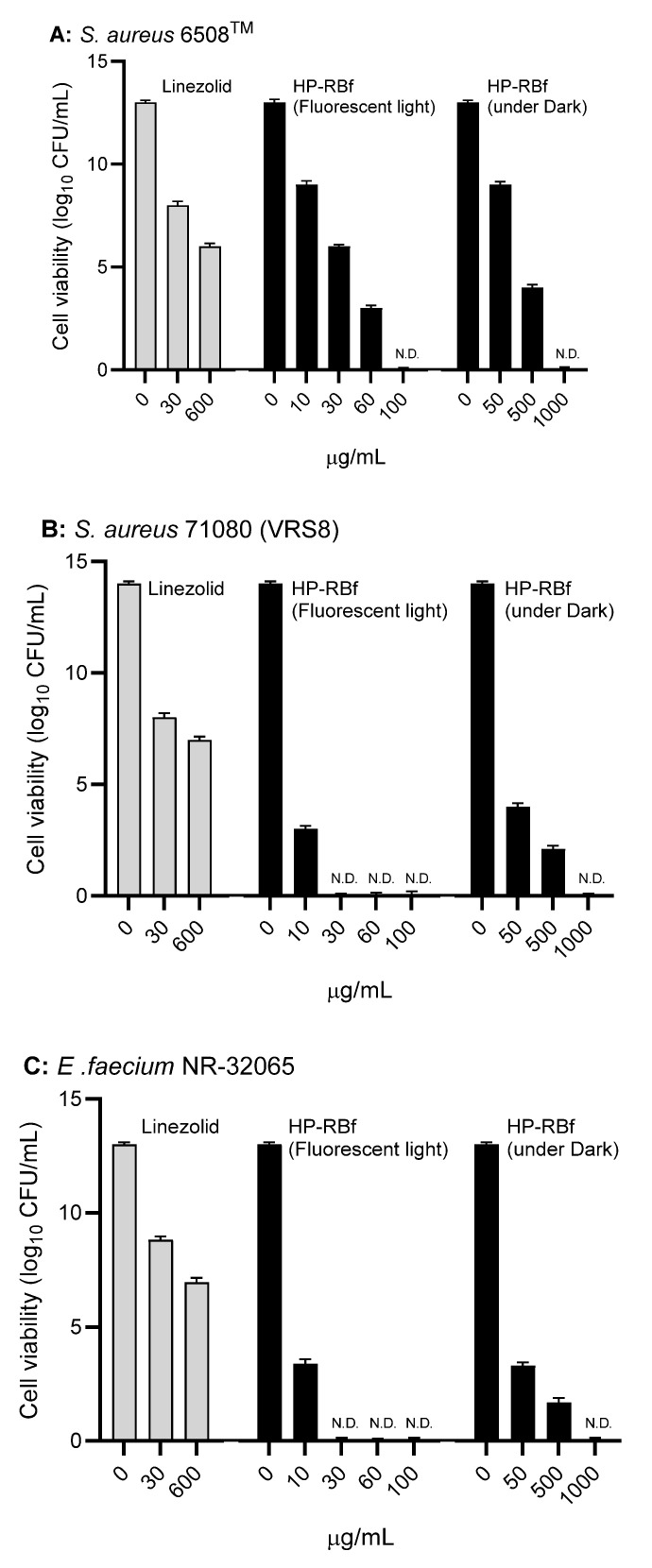
Anti-biofilm activity of HP-RBf against Gram-positive bacteria. (**A**) Anti-biofilm activity of HP-RBf against *S. aureus* 6538^TM^; (**B**) Anti-biofilm activity of HP-RBf against *S. aureus* 71080(VRS8); (**C**) Anti-biofilm activity of HP-RBf against *E. faecium* NR-32065.

**Figure 4 molecules-27-00322-f004:**
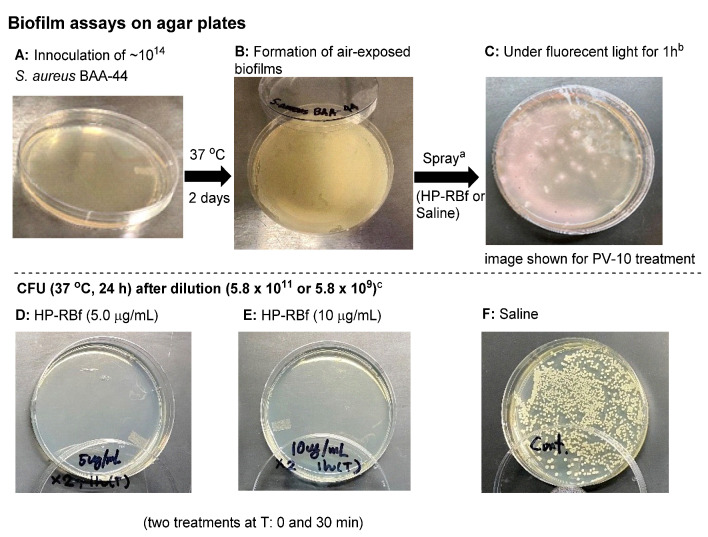
Effect of HP-RBf on air-exposed biofilms of *S. aureus* BAA-44^TM^. ^a^ HP-RBf or saline was treated twice at Time 0 and 30 min (total treatment time: 1 h). ^b^ 17 W, 63.8 cm^2^ fluorescent light was used. ^c^ CFU was counted after a 24 h incubation at 37 °C (see Table 3).

**Figure 5 molecules-27-00322-f005:**
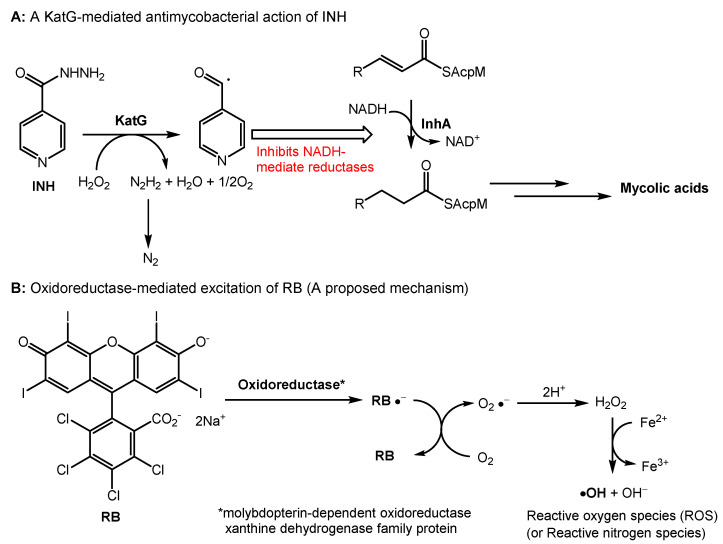
A plausible mechanism of antibacterial activity of RB in dark conditions. (**A**) A mechanism of antimycobacterial activity of INH; (**B**) A proposed mechanism of antibacterial activity of RB in dark conditions.

**Figure 6 molecules-27-00322-f006:**
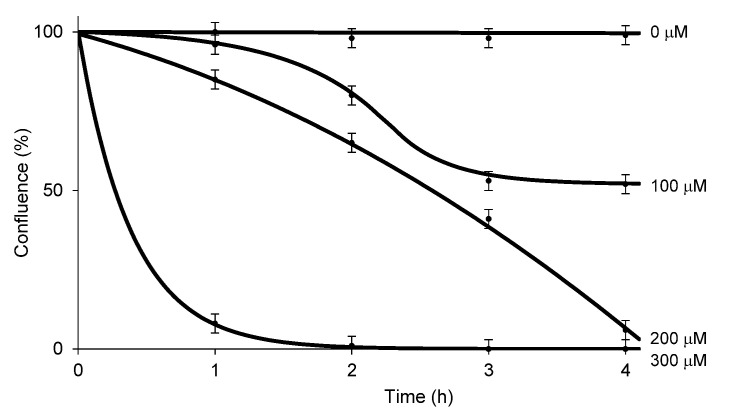
Cell confluence (%) vs. incubation time with HP-RBf in Vero cells under the fluorescent light (17 W, 63.8 cm^2^, 0–1.1 KJ/cm^2^).

**Figure 7 molecules-27-00322-f007:**
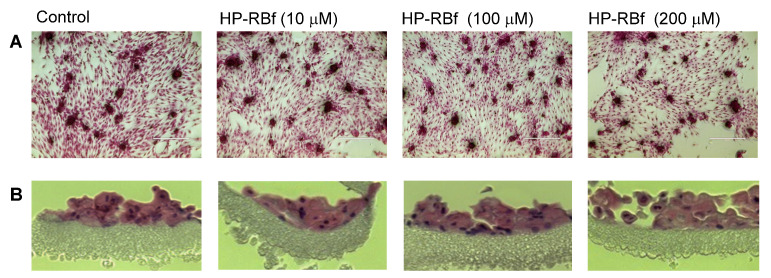
Integrity of multi-layered human epidermal keratinocytes (HEKa) cells by treatment of HP-RBf under the fluorescent light. The cells were grown (25 days) using the Nunc Cell Culture Insert system (CLS-AN-047W) [67]. HEKa cells were treated with HP-RBf (10–200 μM) in the fluorescent light for 1 h (17 W, 63.8 cm^2^, 0.96 KJ/cm^2^). Stained with hematoxylin and eosin; (**A**) top view (×10) (**B**) side view (×40).

**Table 1 molecules-27-00322-t001:** MIC of a series of Gram-positive and -negative bacteria via the broth dilution method ^a^.

Entry	Bacteria ^b^	MIC (μg/mL) ^c^ (Fluorecent)	MIC (μg/mL) ^d^ (LED)	MIC (μg/mL) ^e^ (Sun)	MIC (μg/mL) ^f^ (Dark)
1	*Bacillus subtilis* ATCC6051	0.78	0.78	0.20	50.0
2	*Bacillus cereus* NRRL B-569	0.20	0.20	0.098	50.0
3	*Bacillus cereus* 13061^TM^	0.78	0.78	0.78	50.0
4	*Staphylococcus aureus* 6538^TM^	1.6	1.6	0.78	50.0
5	*Staphylococcus aureus* subsp. *aureus* BAA-1683 (a MRSA strain)	3.1	3.1	- ^g^	50.0
6	*Staphylococcus aureus* subsp. *aureus* BAA-41^TM^ (a MRSA strain)	1.6	0.78	- ^g^	25.0
7	*Staphylococcus aureus* subsp. *aureus* BAA-42^TM^ (a MRSA strain)	1.6	1.6	- ^g^	50.0
8	*Staphylococcus aureus* subsp. *aureus* BAA-44^TM^ (a MRSA strain)	1.6	1.6	- ^g^	50.0
9	*Staphylococcus aureus* subsp. *aureus* BAA-2094^TM^ (a MRSA strain)	1.6	1.6	- ^g^	25.0
10	*Staphylococcus aureus* BAA-2313^TM^ (a MRSA strain)	1.6	0.78	- ^g^	25.0
11	*Staphylococcus aureus* subsp. *aureus* 33592^TM^ (a methicillin and gentamicin resistant strain)	0.78	0.78	- ^g^	50.0
12	*Staphylococcus aureus* AIS2006032 (a vancomycin-resistant strain)	0.78	0.78	- ^g^	50.0
13	*Staphylococcus aureus* BR 5 (A methicillin and vancomycin resistant strain)	0.78	0.78	- ^g^	25.0
14	*Staphylococcus aureus* strain AIS 1,000,505 (VRS10, a vancomycin-resistant strain)	0.78	0.78	- ^g^	25.0
15	*Staphylococcus aureus* USA100 strains 71,080 (VRS8, a vancomycin-resistant strain)	0.39	0.39	- ^g^	25.0
16	*Staphylococcus epidermidis* 35984^TM^	0.78	0.78	0.39	25.0
17	*Enterococcus faecalis* 19433^TM^	0.78	0.39	- ^g^	25.0
18	*Enterococcus faecium* 349^TM^	0.78	0.39	- ^g^	50.0
19	*Enterococcus faecium* BAA-2320	0.78	0.39	- ^g^	25.0
20	*Enterococcus faecium* NR-32065 (a vancomycin-resistant strain)	0.78	0.78	- ^g^	50.0
21	*Enterococcus faecium* patient #3-1, NR-31912 (a vancomycin-resistant strain)	0.78	0.78	0.78	50.0
22	*Enterococcus faecium* UAA714 (a vancomycin-resistant strain)	0.78	0.78	0.78	50.0
23	*Streptococcus salivarius* subsp. *salivarius* 7073™	0.78	0.78	0.39	25.0
24	*Streptococcus pneumoniae* 6301^TM^	50.0	50.0	- ^g^	>100
25	*Mycobacterium smagmatis* 607^TM^	12.5	12.5	- ^g^	25.0
26	*Mycobacterium avium* subsp. *avium* 2285	25.0	25.0	- ^g^	50.0
27	*Mycobacterium kansasii* 824^TM^	25.0	25.0	- ^g^	50.0
28	*Mycobacterium bovis* 35734^TM^ (BCG)	25.0	25.0	- ^g^	50.0
29	*Mycobacteroides abscessus* 19977^TM^	25.0	25.0	- ^g^	50.0
30 ^h^	*Saccharomyces cerevisiae* BY4743	12.5	12.5	12.5	125
31	*Echerichia coli* 35218^TM^	50.0	50.0	50.0	>200
32	*Echerichia coli* TW07793 serotype O157	100	100	- ^g^	>200
33	*Echerichia coli* serotype O157 43888^TM^	50.0	50.0	50.0	>200
34	*Pseudomonas aeruginosa* 27853^TM^	50.0	50.0	- ^g^	>200
35	*Pseudomonas aeruginosa* MRSN 1356	>100	>100	- ^g^	>200
36	*Klebsiella pneumoniae* 8047^TM^	>100	>100	- ^g^	>200
37	*Klebsiella pneumoniae* CHS 67	50	50	- ^g^	>200
38	*Acinetobacter baumannii* 19606^TM^	>100	>100	- ^g^	>200
39	*Acinetobacter baumannii* BAA1800^TM^	>100	>100	- ^g^	>200
40 ^i^	*Bacteroides fragilis* 25285^TM^	>100	>100	- ^g^	>200
41	*Salmonella subsp. enterica* Typhimurium BAA-2721^TM^	12.5	12.5	- ^g^	>200
42	*Salmonella enterica* Pennsylvania Tomato Outbreak, *Serovar* Typhimurium, Isolate 1 NR4333	100	100	- ^g^	>200
43	*Burkholderia multivorans* CGD1	6.25	6.25	- ^g^	>200
44	*Burkholderia cepacia* UCB 717	6.25	6.25	12.5	>200
45	*Burkholderia cepacia* genomovar III LMG 16656	6.25	6.25	- ^g^	>200
46	*Burkholderia cepacia genomovar* VI LMG 18941	6.25	6.25	- ^g^	>200
47	*Proteus mirabilis* urine isolate WGLW4	3.13	3.13	3.13	>200

^a^ All experiments were triplicated. The MIC values were determined via OD and colorimetric assays using risazurin or malachite green; ^b^ Bacteria were purchased from ATCC or acquired from BEI Resources; ^c^ A 17 W, 63.8 cm^2^ fluorescent light was used. The MIC was determined after 24 h of treatment (23.0 KJ/cm^2^); ^d^ A 9.5 W, 28.3 cm^2^ LED light was used. The MIC was determined after 24 h of treatment (29.0 KJ/cm^2^); ^e^ The experiments were performed in the BSL-2 lab on the 5th floor, College of Pharmacy, UTHSC. The 96-well plates were placed on the east side of the lab and exposed to sunlight filtered through an architectural window. The experiments were terminated after 9 h (8 AM-5 PM, sunny, 34 °C (outside), 27 °C (inside)); ^f^ The experiments were performed in the dark room. The 96-well plates were covered with an aluminum foil. The MIC values were determined after 24 h; ^g^ MIC was not determined; ^h^ One yeast strain was examined; ^i^
*Bacteroides fragilis* was grown in an anaerobic chamber under an atmosphere of a mixture of H_2_ and N_2_ (5/95%) with a palladium catalyst.

**Table 2 molecules-27-00322-t002:** Difference in the MIC (agar dilution vs. broth dilution) and MBC of HP-RBf under fluorescent light ^a^.

Entry	Bacteria ^b^	MIC (μg/mL) ^c^ via Agar Dilution	MBC (μg/mL) ^c,d^ via Agar Dilution	MIC (μg/mL) ^c,e^ via Broth Dilution	MIC (μg/mL) ^f^ via Agar Dilution (Dark)
1	*Bacillus subtilis* ATCC6051	0.01	0.02	0.78	125
2	*Bacillus cereus* 13061^TM^	0.01	0.20	0.20	125
3	*Staphylococcus aureus* 6538^TM^	0.05	0.05	1.6	25.0
4	*Staphylococcus aureus subsp. aureus* 33592^TM^ (a methicillin and gentamicin resistant strain)	0.10	0.50	0.78	50.0
5	*Staphylococcus aureus* USA100 strains 71,080 (VRS8, a vancomycin-resistant strain)	0.20	0.50	0.39	50.0
6	*Staphylococcus aureus* subsp. *aureus* BAA-44^TM^ (a MRSA strain)	0.39	0.78	1.6	50.0
7	*Enterococcus faecium* NR-32065 (a vancomycin-resistant strain)	0.20	0.78	0.78	
8	*Echerichia coli* 35218^TM^	6.3	1.6	50.0	>200
9	*Burkholderia cepacia* UCB 717	0.78	3.1	6.3	>100
10	*Mycobacterium smegmatis* 607^TM^	3.1	6.3	12.5	50.0
11	*Saccharomyces cerevisiae* BY4743	1.6	400	12.5	>500

^a^ All experiments were triplicated. The MIC values were determined via counting the colony-forming units (CFUs); ^b^ Bacteria were purchased from ATCC or acquired from BEI Resources; ^c^ A 17 W, 63.8 cm^2^ fluorescent light was used. The MIC was determined after 24 h of treatment (23.0 KJ/cm^2^); ^d^ MBC: minimum bactericidal concentrations (μg/mL); ^e^ See Table 1; ^f^ The experiments were performed in the dark room. HP-RBf-agar prepared in the 24-well plates and 35 mm culture dishes were covered with an aluminum foil.

**Table 3 molecules-27-00322-t003:** CFU after the treatment of HP-RBf for air-exposed biofilms of *S. aureus* BAA-44^TM a^.

Molecule	CFU/mL (dilution 5.8 × 10^11^) ^c^	CFU/mL (dilution 5.8 × 10^9^) ^c^
HP-RBf (5.0 μg/mL) ^b^	0 (0 colony)	9.4 × 10^10^ (35 colonies)
HP-RBf (10 μg/mL) ^b^	0 (0 colony)	4.5 × 10^10^ (17 colonies)
Saline (control) ^b^	5.8 × 10^14^	9.3 × 10^14^

^a^ The procedure is illustrated in Figure 4. ^b^ 17 W, 63.8 cm^2^ fluorescent light was used. ^c^ CFU was counted after a 24 h incubation at 37 °C.

**Table 4 molecules-27-00322-t004:** MICs of HP-RBf and antimycobacterial agents against *M. smegmatis* and its HP-RBf resistant strain ^a^.

Drug	MIC (μg/mL) against *M. smegmatis* 607^TM^ (Wild-Type)	MIC against *M. smegmatis* 607^TM^ (RB Mutant)
HP-RBf	12.5	200
Isoniazid (INH) ^b^	1.56	25–50
Ethionamide (ETH)	12.5	12.5
Amikacin	0.78	0.78
Capreomycin	3.13	3.13
Rifampicin	1.56	1.56
APPB ^c^	0.20	0.20

^a^ The MIC values were determined in dark condition. All experiments were triplicated; ^b^ An inhibitor of mycolic acid synthesis; ^c^ APPB = aminouridyl phenoxypiperidinylbenzyl butanamide, an MraY/WecA inhibitor.

## Data Availability

Not available.

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
