# Peer review of "Antibacterial Activity of Pharmaceutical-Grade Rose Bengal: An Application of a Synthetic Dye in Antibacterial Therapies"

_molecules, 2022, doi:10.3390/molecules27010322_

Round 1

Reviewer 1 Report

The authors considered most of the reviewer's comments in the submitted manuscript. Some minor additions are still required, mainly in the references section.

Line 427 – Complete the references section with the CLSI entry.

Lines 622, 634 - References 19 and 25 refer to the same publication - delete one of the points.

Author Response

Rev 1:
Line 427 – Complete the references section with the CLSI entry.
The line # is not match with the original manuscript. But, we have corrected as much as we could
figure out.
The reference was included.

Lines 622, 634 - References 19 and 25 refer to the same publication - delete one of the points.
The duplication was fixed. Referenced 25 was swapped with the other article.

Reviewer 2 Report

The document is a revised manuscript on the use of Rose Bengal dye as antibacterial.

Many of the suggestions have been incorporated, although I consider there are still some problems to be corrected. 

The introduction is too large, with information that is not related to the main objective of the manuscript. There is a lack of related research on the antimicrobial and antiproliferative activity of the dye. 

The conclusion also needs to be more precise on the main findings. The objective described does not describe the information that is included in the manuscript.

In line 409, please delete the repeated "described"
In line 419, please correct the incubation period

references for section 3.6, 3.7, 3.9, 3.10 are still missing

There are still incorrect identification of scientific names v.gr. line 525

Author Response

The introduction is too large, with information that is not related to the main objective of the
manuscript. There is a lack of related research on the antimicrobial and antiproliferative activity of
the dye.
The introduction is shortened; the redundant portions are deleted. This article describes, for the
first time, that a pure grade of rose bengal’s antibacterial activity is summarized comprehensively.

The conclusion also needs to be more precise on the main findings. The objective described does
not describe the information that is included in the manuscript.
Conclusion is slightly revised. The reviewer may miss the significance of our studies. The
important discussion in Conclusion is highlighted in red. The companies cannot develop rose
benagal to improve human health because pharmaceutically relevant rose bengal was not available.
The biological data generated with crude rose bengal are also not useful to move forward to
clinical studies.

In line 409, please delete the repeated "described"
Corrected.

In line 419, please correct the incubation period.
Revised.

References for section 3.6, 3.7, 3.9, 3.10 are still missing.
Appropriate references are included for these sections.

There are still incorrect identification of scientific names v.gr. line 525.
The line # is not exactly match with the original manuscript. We have checked the terms and names
used in this manuscripts. All bacterial names were also checked. 

Round 2

Reviewer 2 Report

No further comments